# Compounding impacts of COVID-19, cyclone and price crash on vanilla farmers' food security and natural resource use

Henintsoa Rakoto Harison[1], James P. Herrera[2,3], O. Sarobidy Rakotonarivo[1]*

1 École Supérieure des Sciences Agronomiques, University of Antananarivo, Antananarivo, Madagascar,
2 Duke Lemur Center SAVA Conservation, Duke University, Durham, NC, United States of America,
3 Evolutionary Anthropology, Duke Global Health Institute, Duke University, Durham, NC, United States of America

* sarobidy.rakotonarivo@gmail.com

**Data Availability Statement:** Interview transcripts cannot be shared publicly because our ethical approval under Boston University's research ethics policy does not allow for sharing of the transcripts

## Abstract

The intersection of the COVID-19 pandemic with other crises can amplify vulnerabilities and push communities further into poverty. In low-income countries, the dual impacts of COVID-19 and extreme weather events, along with multidimensional poverty and structural vulnerabilities in agriculture can decimate farmer livelihoods. This study aims to understand the effects of individual and compounding crises (COVID-19, cyclones, and vanilla price collapse) on smallholder vanilla farmers and local coping strategies in Madagascar, one of the world's largest vanilla producers and poorest countries. We used semi-structured and scenario-based interviews across two case study villages with contrasting enforcement of forest regulations. We found that the impact of the pandemic, combined with the cyclone event, disrupted livelihoods, resulting in income losses and food security challenges that exacerbated farmer vulnerabilities. Sixty eight percent of households reported crop losses due to strong winds and heavy rainfall brought by cyclone Enawo in 2017. The COVID-19 outbreak struck the region just as the residents were recovering from the effects of the cyclone. COVID-19-related travel restrictions in the aftermath of the cyclone took a substantial economic toll, with 54.1% of respondents experiencing a decline in earnings, and 17% facing a total loss of income due to the imposed lockdown. The decline in vanilla prices at the onset of 2020 had a far-reaching additional impact, affecting not only farmers but also residents who rely on other sources of income. Local communities reported using the forest resources more frequently as a safety net during crises in the village with more lenient regulations. This study underscores the importance of understanding the interconnectedness and compounding impacts of cascading crises on food security and natural resource use. We highlight the need for a comprehensive approach to increasing farmer resilience, particularly for those reliant on global market crops such as vanilla.

to anyone outside of the research team, and the interview transcripts cannot be sufficiently anonymized to prevent deductive disclosure. Any researcher who would like to access these confidential data transcripts (in Malagasy) would need to make a specific request to Shayne C. Deal, the Analyst/Reliance Agreement Specialist Of Boston University Institutional Review Board (Sdeal101@bu.edu).

**Funding:** This work was funded by the USAID Partnerships for Enhanced Engagement in Research (PEER) programme (AID-OAA-A-11-00012 awarded to OSR). OSR was also funded by the European Union (Grant no. DCI-PANAF/2020/420-028), through the African Research Initiative for Scientific Excellence (ARISE) pilot programme. ARISE is implemented by the African Academy of Sciences with support from the European Commission and the African Union Commission. The funders had no role in study design, data collection and analysis, decision to publish, or preparation of the manuscript.

**Competing interests:** The authors have declared that no competing interests exist.

## Introduction

Global economies are increasingly vulnerable to a confluence of crises, spanning pandemics such as COVID-19, economic shocks, and extreme climatic events [1]. Extreme weather events can lead to significant adverse societal costs [2, 3]. They can pose significant challenges for various agricultural and industrial sectors, and disrupt global supply chains and the sustainable development prospects of the most vulnerable nations [4–6]. The COVID-19 pandemic emerged into an already complex context of a rapidly changing climate exacerbating existing vulnerabilities [7, 8]. Other economic shocks can further aggravate the dual threat of extreme weather events and the pandemic on global and regional economies [9].

The intersection of the COVID-19 pandemic with other crises can amplify vulnerabilities and push communities further into poverty. The repercussions are particularly severe in low-income countries, where limited resilience and adaptive capacities can amplify the social impacts of compounding crises and heighten vulnerabilities [10]. Although these compounding effects are acknowledged, most studies report the impacts of specific crisis separately, and we lack a more in-depth understanding of the breadth and magnitude of the compounding impacts of various crises on farmer vulnerability.

Amidst this global backdrop, Madagascar emerges as a critical case study, epitomizing the challenges faced by one of the world's poorest nations [11]. With an economy heavily reliant on agriculture and burdened by high poverty rates, Madagascar exemplifies the intricate interactions between global, regional, and local crises [12]. Madagascar is an important case study for how compounding crises such as the COVID-19 pandemic, extreme weather events and market price volatility impact smallholder farmers who are already facing challenges of climate change and food insecurity.

The vanilla sector plays a major role in Madagascar's economic fabric [13]. The SAVA region in northeast Madagascar is the country's main producer of vanilla with an estimated 70,000 farmers in 2018 producing nearly 70–80% of the global Bourbon vanilla [13]. Prior to the pandemic in 2017, the SAVA region was severely affected by a devastating cyclone, Enawo, which destroyed farms, including vanilla plantations. The COVID-19 outbreak struck the region just as the residents were recovering from the effects of cyclone Enawo and further amplified vulnerabilities. The collapse in the vanilla market prices at the onset of 2020 can significantly exacerbate these impacts.

This research aims to study the individual and compounding effects of global crises (COVID-19, cyclones and vanilla price collapse) on farmers' lives across two case study villages in northeast Madagascar. Using semi-structured and scenario-based interviews, we identified various mechanisms by which these cascading crises can consecutively and concurrently impact the livelihoods and food security of farmers in these case villages. We further explored the knock-on impacts on natural resource use and biodiversity. Finally, we report the various strategies employed by farmers to cope with these challenges. By focusing on the context-based specific realities of the case study villages, this article contributes to a better understanding of the challenges faced by farmers in this region of Madagascar and provides wider lessons for similar contexts in low-income countries. We explored these research aims using a qualitative approach. Qualitative approaches can help provide detailed, vivid, and context-sensitive descriptions of the data and gain an in-depth understanding of the context-specific challenges and nuanced perspectives of place-based actors [14].

## Methods

The study was conducted in two rural villages in the SAVA region, Mandena and Andrapengy. Mandena is a village located between the cities of Andapa and Sambava along the national

road 3b but approximately 5km off the paved road on a dirt path. Mandena is directly adjacent to the entry of Marojejy National Park, where forest protection has relatively been heavily enforced since 1950 [15]. The village is divided into four sectors, with a total of 547 households, and an average of 137 households per sector. Andrapengy is situated between the towns of Antalaha and Sambava, along the national road 5a and is more accessible than Mandena by virtue of being directly on the paved road. Unlike Mandena, which is situated adjacent to the Marojejy National Park, Andrapengy does not have a national park, though remnant patches of forest exist in the proximity of the village. The village of Andrapengy is divided into six sectors, with a total of 265 households, and an average of 44 households per sector.

The data collected for this study were qualitative in nature and obtained through semi-structured interviews and scenario-based interviews. The semi-structured interview guides (S1 File) consisted of 18 questions that covered participant basic demographics, including age, household size, education level, and income sources. These interviews also examined the impacts of various longitudinal crises, such as cyclones in the past, the COVID-19 pandemic from 2020–2022, and fluctuations in vanilla prices, dating back to the early 2000s, on livelihoods and food security. Additionally, participants were asked about the coping strategies they adopted in response to each crisis and the potential effects of each crisis and their compounding effects on the use of natural resources.

The scenario-based interview guide (S2 File) covered six hypothetical scenarios encompassing the status quo, an increase or decrease in vanilla prices, land degradation, diseases affecting vanilla cultivation, the restrictions during COVID-19, and extreme weather events such as cyclones. For each scenario, participants were prompted to share their anticipated effects on their lives, and their coping strategies. This method allowed us to delve into the nuanced decision-making processes of farmers, including a long-term perspective on adaptive responses they have used and their perspectives in the face of uncertainty. Respondents were prompted to speak on any of the crises they chose to discuss, without restricting the timeframe.

We first checked the sample frame (the full household listing in each sector provided by local authorities) for any omission with key-informants in the village (including the village head, and local guides) identified with the assistance of villagers. We then randomly selected 30 participants per village, ensuring representation from each sector of each village. In total, we conducted semi-structured interviews and scenario-based interviews with a total of 60 participants in the two villages.

The interviews were conducted in January and February 2023 after we obtained an ethical clearance from Boston University (IRB Protocol 6934X). According to our approved protocol, respondents gave their verbal consent after they were informed about the aim of the study, the potential use of the datasets, that the data would remain anonymous with only researchers on the team having access to the transcripts, and were told that they could elect not to answer any questions the wished, or stop the interview anytime.

We analyzed the data using thematic and content analysis methods [16, 17]. Thematic analysis involved identifying and analyzing themes in the data, including initial coding of meaningful units, searching for common themes, revising and defining themes, and interpreting the results [16]. We specifically searched for any themes and patterns indicating the potential effects of various crises, including COVID-19, cyclones, and vanilla price fluctuations on household livelihoods and food insecurity, the effects of these crises on the use of natural resources and biodiversity, and the diverse strategies adopted by farmers. The aim is to uncover and understand key themes in the data and how they relate to one another. The content analysis involved a systematic analysis of the key themes identified from the thematic analysis [17]. It focuses on the reoccurrence of themes at a more surface-level of analysis i.e. their frequency.

Researchers and readers should use caution when interpreting such frequencies, however [18]. Frequencies may help get a better feel for how often themes occur within the specific sample, and advance qualitative research within a discipline that are not familiar with or oriented to qualitative research. Nevertheless, incorporating frequencies in qualitative research may misrepresent data as not every interviewee is asked the same questions in the same way [19]. Number can also detract from the more valuable, detailed, and nuanced insights uncovered from qualitative research [19]. Similarly, though a theme or response that only occurs once may seem unimportant based on frequency of occurrence, it may represent the unique challenge, solution, idea, or theory that could influence social, anthropological, and policy change. Lastly, readers should not interpret frequencies as generalizations to broader populations; instead, one should assess the transferability of the findings, sometimes also referred to as analytic generalization or case-to-case generalization [18, 19].

Coding was conducted iteratively by HRH to generate codes and themes that summarized and captured the essence of the responses. a second coder (OSR) used the draft coding scheme to independently code five randomly selected sample transcripts. To ensure consistency in coding and mitigate the subjectivity of a single coder, we calculated the intercoder reliability, the Cohen's Kappa coefficient which is a numerical measure of the agreement between different coders regarding how the same data should be coded [20]. We considered a coefficient with a value of 0.70 or higher acceptable. Codes that did not meet this threshold were jointly reviewed, and conflicting interpretations were resolved by either merging codes or refining code definitions. These analyses were conducted with NVivo 12 software.

## Results

### Participant characteristics

In the interviewed sample, 40% were women. Almost all (90%) interviewees identified themselves as farmers, with agriculture as their primary source of income. Among the total participants, 97% were involved in rice cultivation, with this staple crop primarily designated for their own consumption, regardless of their primary occupation. Within the group of farmers, 75% were engaged in vanilla farming, and for 67% of these individuals, vanilla cultivation served as their primary source of income. Additionally, 53% of all participants reported having a secondary source of income, with the majority working as farm laborers (47%), followed by charcoal production (16%). On average, interviewees were 49 years old, had completed five years of education, and possessed one hectare of arable land on average (S1 Table).

### COVID-19 disrupted travel and undermined local livelihoods

The restrictions imposed by the Malagasy government to contain the spread of COVID-19 had significant consequences on the livelihoods and food security of the farming community (S2 Table presents an overview of these social impacts and local strategies, the various themes reported, and frequency of these themes).

The measures implemented, such as quarantine and travel restrictions, prevented farmers from selling their products in other cities. As a result, the majority of households reported their income was impacted by the pandemic. Approximately 54.1% of households reported they experienced a decline in their earnings, while 17% suffered a total loss of income due to the imposed lockdown.

In addition to the financial challenges resulting from reduced income, farmers also faced difficulties caused by the pandemic itself. Four households reported that they contracted the virus, rendering them unable to work and provide for their families. Consequently, these

households had to reduce both the quantity and quality of their food intake, leading to a deprivation of essential nutrients.

Furthermore, the burden of medical expenses related to COVID-19 added to the financial hardships of households. The four households who reported contracting the virus reported they had to allocate a significant portion of their income to cover healthcare costs, further exacerbating their already precarious situation. Moreover, farmers with the means to hire wage laborers reduced wages as a means of coping with the economic consequences of the pandemic.

In addition to the impacts of the disease and healthcare, the challenges faced by farmers during this time were particularly striking. The restrictions imposed to curb the spread of the virus resulted in a substantial decrease in the prices of agricultural goods. These restrictions prevented farmers from freely selling their produce, leading to limited market access and diminished income.

As one farmer explained,

*"Overall, the restrictions resulting from the pandemic had a detrimental effect on the prices of goods, especially agricultural crops. Agricultural products were sold at prices far below their prices prior to the pandemic. For instance, let's consider the case of pineapples, which used to be sold for 5000 Ariary, but were sold for only 1000 Ariary during the height of the crisis. This significant drop in agricultural product prices has had a profound impact on our ability to purchase essential goods."* (Male, 59 years old, Andrapengy).

For vanilla farmers, the situation was exasperating due to the travel restrictions. Among the households that suffered income losses during the pandemic, those who rely on vanilla cultivation as their primary means of subsistence were particularly affected. The prohibition on travel and transportation between different regions, districts, and even cities disrupted market dynamics. Buyers were unable to travel to the villages where vanilla is typically sourced, while, concomitantly, sellers were unable to reach or connect with potential buyers. Approximately 70% of the households, predominantly farmers, found themselves in a dire situation. One vanilla farmer shared her experience, saying,

*"The impact was substantial. During these periods of health emergency, no vanilla was sold because we had no buyers. We faced tremendous difficulties, and our means of subsistence were severely compromised."* (Female, 48 years old, Mandena).

The repercussions of COVID-19 were not limited to the agricultural sector alone. The service sector also experienced a substantial negative impact, with around 7% of individuals (carpenters, grocers, small food vendors, laborers, etc.) witnessing a 50% decrease in their income. One resident, who runs a grocery shop, shared their personal experience, stating,

*"We faced severe financial crises within our household due to the numerous constraints imposed by the pandemic. We were unable to work due to the restrictions, and alternative options were scarce. The decrease in demand for our services has greatly affected our livelihoods. Previously, we used to earn 10,000 Ar per day, but now we can only generate a turnover of 5,000 Ar per day."* (Male, 52 years old, Andrapengy.)

Additionally, the food security of farmers was compromised. Approximately 46% of households reported significant changes in their dietary patterns. A marked reduction in the consumption of staple foods, particularly rice, was observed among the majority of households.

Three households (5%) were compelled to entirely eliminate rice from their meals due to a decline in income, further aggravated by soaring food prices.

A compelling case illustrating this situation was reported in Andrapengy village, where a family recounted their experience:

*"There has been a considerable decrease in the amount of rice consumed. Previously, we used to consume 4 cups of rice around the same time of the year, but we had to reduce it to 2.5 cups once the COVID-19 pandemic struck. . . At times, we even reduce the frequency of meals, resulting in only the children having a meal in the evening while the adults go to bed on an empty stomach."* (Male, 41 years old, Andrapengy)

## Colliding crises amplified vulnerabilities in the wake of COVID-19

The COVID-19 pandemic arrived at an unfortunate time, exacerbating the already precarious situation in the region. COVID-19 outbreak struck the region just as the residents were recovering from the effects of the cyclone Enawo. Cyclones, such as the significant Enawo in 2017, are a major threat to the region annually. The overall dependence of farmers in the region on vanilla also makes them vulnerable to challenges of extreme price volatility. These additional crises interact to exacerbate the vulnerability of these marginalized communities.

The 2017 Enawo cyclone's impact caused significant devastation in the two villages, resulting in increased vulnerability due primarily to crop losses. Among the surveyed households, 68% reported crop losses, particularly essential crops like vanilla and cloves, due to strong winds and heavy rainfall. Furthermore, the destruction of homes left 21 out of 60 households homeless, leading to social instability as they sought new places to live or relied on the support of nearby families. One resident shared the psychological distress caused by the cyclone, stating:

*"My house was ravaged, my vanilla crops were destroyed, and the bananas, which were also devastated, were not yet available or ripe. It almost drove me insane."* (Male, 53 years old, Mandena)

On the food front, the situation deteriorated with crop losses and decreased agricultural incomes, resulting in severe food security challenges. Among the interviewed households, 67% reported changes in their dietary habits, mainly experiencing a reduction in rice consumption, and it was found that approximately 3% of households shifted from eating three meals a day to two. The destruction of agricultural infrastructure further compromised food security, as it had a significant impact on rice production. Almost all (97%) of households cultivated rice, with approximately 77% being exclusively for subsistence. The cyclone's impact led to the inundation of a significant portion of rice crops, resulting in reduced harvests and diminished availability for consumption.

The situation was particularly dire in the village of Andrapengy, which is located within kilometers of the Indian Ocean. The loss of homes and income due to the cyclone drove most villagers to urgently seek construction materials for rebuilding their destroyed houses, leading them to rely on forest resources. Both local residents and migrants turned to using wood from the nearby forests on the mountain known locally as Ambanitaza to produce charcoal and wooden planks. This increased dependence on forest resources resulted in overexploitation, depleting the available wood supply. The village chief explained:

*"We had a large reserve [classified forest] in Ambanitaza before the cyclone, but it was ravaged. The inhabitants then exploited the parts of the reserve devastated by the cyclone to obtain wood, especially for making planks. However, they also used the reserve for cooking,*

*inadvertently triggering a bushfire that destroyed the reserve. Subsequently, we faced significant difficulties in finding wood to rebuild and renovate our houses."* (Male, 59 years old, Andrapengy)

Moreover, just when the residents thought they might find some respite after the successive shocks of the cyclone and COVID-19, the situation further deteriorated due to fluctuations in vanilla prices. During the pandemic, demand for vanilla decreased, partly due to the absence of buyers because of the travel restrictions. The lack of buyers amplified the negative effects of the decline in vanilla prices, further exacerbating the struggles of farmers. For instance, in 2018, vanilla prices ranged from $600 to $750 per kilogram, but by 2020, they plummeted to just $110 per kilogram, and this downward trend continues [21].

Interviews conducted among the local population further highlighted the magnitude of the impact of declining vanilla prices. Out of all respondents, 87% experienced a decrease in their incomes and 10% faced a total loss of cash income, exacerbating their already precarious economic conditions. Furthermore, due to financial constraints, 62% of the respondents were compelled to reduce their food consumption, with rice being the primary staple affected.

Those farmers who were not primarily reliant on vanilla as a source of income (25%) still faced challenges due to the vanilla price crash because it caused the price of other crops to drop as well. Notably, the respondents highlighted that the local economy relies on vanilla farmers, who act as potential buyers in the village. As a consequence, if vanilla prices decline, it significantly impairs the purchasing power of these farmers. Consequently, farmers selling other crops were compelled to decrease their prices in order to generate any revenue, as not doing so would lead to no income at all. One farmer expressed the situation:

*"The decline in vanilla prices had a profound impact on me and my family. Indeed, when the price decreased, we faced considerable financial challenges, as it affected not only us but also other community members, particularly the vanilla farmers, who were also experiencing financial strain. As a result, all the crops we farm and sell fetched significantly lower prices."* (Male, 25 years old Andrapengy)

In parallel, households dependent on income from daily wage labour faced income loss due to a decrease in job opportunities. Approximately 3% of respondents rely on daily wage labor as their main source of income and 18% report it as their secondary source. When vanilla prices were high, some vanilla farmers could hire other individuals for farming their land, tending to their rice fields, and assisting with various agricultural activities. However, with the income challenges brought on by the decline in vanilla prices, they had to reduce the number of farm laborers they employed or take on the tasks themselves. Two farmers shared their experiences, illustrating the difficulties they encountered. A vanilla farmer who used to employ people for farming activities, expressed,

*"Since the vanilla price declined, we have faced problems with our livelihoods. Previously, we were accustomed to hiring people for various farming tasks, including our crops and rice fields. But now, our earnings no longer permit us to hire laborers, and we are left with the burden of carrying out all these tasks by ourselves, which is proving to be quite challenging."* (Male, 57 years old, Mandena)

One farm worker added: *"All of us were in a difficult situation; we struggled to find jobs since the main employers, the vanilla farmers themselves, were also facing difficult circumstances."* (Female, 60 years old, Andrapengy)

## Local strategies: Coping with income losses and increasing food insecurity

Diversification of crops, particularly fast-growing annual subsistence crops, has emerged as the primary strategy adopted by households to mitigate income losses during the various crises. Consequently, the produce from these crops can be sold in local markets or used for household consumption. Interview data revealed that during each past crisis, most households adopted this diversification strategy, with the percentage of adoption varying with the specific crisis. In the aftermath of COVID19 and the cyclones, it was adopted by at least 50% of households. During the prolonged declines in vanilla prices, up to 70% of residents opted to diversify their crops for selling purposes. Engaging in daily wage labor emerged as the second most used strategy. This was followed by making budget adjustments. Some households (11%) had to lower their standard of living or reduce expenses related to children's education. Additionally, they curtailed their food expenditures to essentials. Furthermore, in certain cases, households resorted to peer-to-peer lending, while others sought loans from associations such as the local Village Savings and Loan Associations.

The interviews suggested that when facing prolonged decline in vanilla prices, cultivating perennial crops such as coffee and cloves emerges as the third most used strategy (13% of the respondents). These crops are cultivated to serve as additional sources of income alongside vanilla, given their favorable market prices and benefits as perceived by farmers. This diversification strategy not only provides a buffer against market volatility but also contributes to the overall resilience of households in managing economic uncertainties.

Given the constant challenges posed by climate change, adopting agricultural practices that are responsive to the local climate conditions was seen by 27 households as a means to enhance resilience. Farmers strategically cultivated crops suitable for dry periods, such as bananas, and adjusted their farming activities to align with rainy and dry periods, including planting leguminous plants (beans) or potatoes. This approach allowed them to optimize yields and ensure food security during varying weather patterns. Additionally, farmers started planting more cassava and yams for consumption, as these crops are suitable for different weather conditions, increasing their resilience. However, amidst the array of modern techniques available, that farmers still exhibit a propensity for traditional methodologies passed down through generations.

## Understanding village dynamics: Mandena and Andrapengy villagers' responses to the crises

The two study communities differed in terms of geographic location and accessibility, impact of the crises, and coping strategies. Mandena is situated near a protected area with relatively more stringent regulations on forest access and use, while Andrapeny, located along a national road, has no such protected area.

At Mandena, located near protected area Marojejy National Park, no anthropogenic activities were reported in the interviews, such as making charcoal or cutting trees for wood planks Approximately 75% of the people interviewed believed that regulations were stringent, with exploiting forest resources strongly prohibited. During the pandemic, despite households (54%) facing financial issues and challenges in dietary habits (15%), their reported strategies focused mainly on crop diversification (both crop sales and subsistence crops). This approach was the most reported by households to cope with income losses (19%). Additionally, some households (4%) resorted to peer-to-peer loans, while others adjusted their budgets to navigate the economic challenges brought on by the pandemic.

Moreover, around 10% of households interviewed in Mandena found resilience through their involvement with the Marojejy National Park. Despite travel restrictions hindering tourism revenue, these households were tasked with patrolling the park and making reports for

foreign organizations during the pandemic. This enabled them to maintain a stable income even when visitors were absent. One respondent shared their experience:

> *"During the COVID-19 pandemic, our financial situation was somewhat moderate, not too critical. We heavily relied on my husband's second source of income as a local guide in the National Park. Even though no visitors came, foreign organizations assigned him other tasks that allowed us to earn an income."* (Female, 42 years old, Mandena)

In contrast, the patterns observed in the village of Andrapengy were different. Andrapengy is located near fragmented and transformed forests known as Ambanitaza, which is not a protected area. Unlike Mandena, the regulations on entry and usage of natural resources in Andrapengy were more lenient. As a result, one common strategy was the use of natural resources, out of 30 respondents, 22 reported exploitation of forest resources during crises. During cyclone periods, the government gave residents permission to access the Ambanitaza forests to collect wood and trees damaged by the cyclone. Consequently, some households (three admitted to doing so, while 22 households reported that exploiting forest resources was common) entered the forests to make charcoal and boards from the trees. This practice became widespread in the village, and charcoal became a prominent strategy for combating crises such as the COVID-19 pandemic, vanilla price decreases, etc. A villager shares their perspective on accessing natural resources:

> *"Charcoal has become a second income-generating activity besides vanilla, and the situation has not improved, given the decline in vanilla prices. . . To this day, charcoal has become a first resort, and most people are doing it; some even rent cars to transport their charcoal from here to Antalaha. . . ."* (Female, 73 years old, Andrapengy).

## Discussion

This study sheds light on significant socioeconomic vulnerabilities faced by farmers in northeastern Madagascar due to the COVID-19 pandemic, as well as the effects of cyclones, and the collapse of vanilla prices. The impacts of government-imposed lockdowns and restrictions on livelihoods were severe, affecting nearly 70% of households. Similar magnitudes of impact were observed in other parts of Madagascar [22] as well as among Ugandan households, where two-thirds reported income losses during the pandemic [23]. Farmers encountered challenges due to travel restrictions that hindered their ability to sell produce freely, resulting in limited market access and reduced income [23].

Household food security was compromised as well, with nearly half of the interviewees reporting changes in dietary patterns and reduced food consumption as coping strategies. Similar coping strategies were documented in Nigeria, where households adjusted their food patterns [24] or increased self-consumption crops, reduced market-oriented production, and enhanced value through home processing and storage in response to the pandemic [25].

Additionally, budget management, daily wage labor, and participation in peer-to-peer lending and Village Savings and Loan Associations were common coping strategies. Existing literature supports the effectiveness of such community-based financial groups during crises. Kansiime et al. (2021) [23] reported that membership in such community-based financial groups, including VSLAs, was associated with an 8% reduction in the likelihood of respondents' income being impacted by the pandemic. This correlation underscores the capacity of these groups to facilitate borrowing for consumption smoothing, enhancing households' ability to cope with income shocks during challenging periods.

In Andrapengy, natural resource use was also reported as a key coping strategy. This was possible due to relatively lenient forest access regulations in the village. In contrast, Mandena reported no natural resources due to strict rules on forest access and use. Nevertheless, this might also suggest that households who live near protected areas might be less willing to report illegal activities [26]. As these households live within a closely-knit community, they might hesitate to report these sensitive activities for fears of potential reprisals. They also claimed to be primarily concerned with their household welfare, and not the businesses of others.

The Marojejy National Park is recognized locally for its stringent set of restrictions and prohibitions on activities ranging from timber extraction to hunting and crop cultivation within the park. These restrictions may have created deeply entrenched local taboos that serve as a partial deterrent from exploiting natural resources in the park. In Uli, Nigeria, traditional taboo practices and traditional belief systems equally encouraged the preservation of tree species, forests and streams [27]. The discrepancy in forest use between the two villages underscores the importance of taboos and social norms in conservation [28] and the need to consider local customs and practices in intervention design.

This study sheds light on the compounding effects of a series of crises on vulnerable populations. Simultaneous shocks can magnify vulnerabilities, leading to cascading impacts on livelihoods and well-being. These findings are in line with previous studies reporting that the pandemic posed new challenges to livelihood strategies while complicating disaster management during cyclone season [29, 30]. The intersection of the pandemic and cyclone Amphan in Bangladesh highlighted the need to consider complex temporalities of disasters [31]. These findings are also consistent with similar studies documenting the compounding impacts of multiple crises on vulnerable populations [32, 33].

The volatility of the vanilla industry significantly worsened the situation for farmers in northeast Madagascar. The COVID-19 pandemic exacerbated the impacts of the decline in vanilla prices due to the absence of buyers and a decrease in global demand. Cyclone Enawo also disrupted the vanilla market prices by contributing to the soaring prices in 2018 and the falling prices at the onset of the COVID-19 pandemic [34]. Historical patterns confirm these trends. For instance, in 2000, Cyclone Hudah passed over the SAVA region, devastating vanilla crops and causing Madagascar's vanilla production to plummet to only 600 tons [35]. Post-Hudah, the global supply of vanilla dropped, causing prices to surge. However, the rising cost prompted many importers to shift to the more affordable synthetic vanillin, resulting in decreased demand. The market became oversaturated, leading to a price crash in 2004, with prices plummeting [35]. This historical instability in the vanilla market negatively impacts the local economy [36].

This study underscores the importance of understanding the interconnectedness of various crises and their compounding impacts on livelihoods, food security, and natural resource use. It provides insights into vulnerabilities faced by households in north-eastern Madagascar and the complex interplay of crises. The findings emphasize the necessity of a comprehensive approach to address challenges arising from the COVID-19 pandemic, cyclones, and economic fluctuations, particularly for those reliant on market crops like vanilla [34]. Heavy dependence on vanilla as a cash crop renders households more susceptible to economic shocks, underscoring the importance of diversifying income sources [37] and promoting resilient agricultural practices [38]. The study also highlights the significance of socio-cultural and ecological contexts when analyzing how farmers cope with various crises.

## Policy implications

Given the relatively small sample and our qualitative approaches, we cannot provide generalizations to the population from which the sample was drawn. Our research findings can

however be transferable to other settings and provide wider lessons for similar contexts in low-income countries. Our study warrants the need for targeted support systems that integrate disaster risk reduction, livelihood diversification, and market resilience strategies to foster long-term sustainability and strengthen the resilience of households facing multiple crises.

Given the vulnerability of households heavily dependent on single cash crops like vanilla, policymakers should promote diversification of income sources and agricultural practices as recommended in other similar contexts [39, 40]. In addition, disaster preparedness and risk reduction efforts can be strengthened through policy interventions. Investing in early warning systems, promoting climate-resilient infrastructure, and fostering community-based disaster management plans can enhance local communities' ability to cope with future cyclones and other environmental challenges [41, 42].

A key aspect of enhancing resilience lies in empowering farmers through agricultural extension services. By disseminating knowledge on improved farming practices, sustainable land management, and climate-smart techniques, farmers can better adapt to changing environmental conditions and build resilience in their agricultural practices [43, 44]. Lastly, fostering cross-sectoral collaboration involving diverse government agencies and actors is important as the impacts of the crises were felt across sectors. It is also essential to involve the local community in the design of policies to understand their needs and increase ownership [45]. Future research avenues could include quantitative estimation of the social impacts of various crises, and investigation of the long-term viability of key policy interventions aimed at increasing local resilience.

## Supporting information

**S1 File. Semi-structured interview guide.**
(DOCX)

**S2 File. Scenario-based interview guide.**
(DOCX)

**S1 Table. Basic demographic information.**
(DOCX)

**S2 Table. Overview of the social impacts of the crises and local coping strategies and frequency of themes.**
(DOCX)

## Acknowledgments

We thank the villagers of Mandena and Andrapengy as well as Randall Kramer, Alexander Duthie and Andrew Bell for comments on an early draft of this manuscript.

## Author Contributions

**Conceptualization:** Henintsoa Rakoto Harison, James P. Herrera, O. Sarobidy Rakotonarivo.

**Formal analysis:** Henintsoa Rakoto Harison.

**Funding acquisition:** O. Sarobidy Rakotonarivo.

**Investigation:** Henintsoa Rakoto Harison, O. Sarobidy Rakotonarivo.

**Methodology:** Henintsoa Rakoto Harison, James P. Herrera, O. Sarobidy Rakotonarivo.

**Project administration:** O. Sarobidy Rakotonarivo.

**Resources:** James P. Herrera.

**Supervision:** O. Sarobidy Rakotonarivo.

**Validation:** O. Sarobidy Rakotonarivo.

**Visualization:** Henintsoa Rakoto Harison.

**Writing – original draft:** Henintsoa Rakoto Harison.

**Writing – review & editing:** Henintsoa Rakoto Harison, James P. Herrera, O. Sarobidy Rakotonarivo.

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
