## [Decision Letter · Decision Letter 0]

28 May 2024

PONE-D-24-09752Compounding impacts of COVID-19, cyclone and price crash on vanilla farmers’ food security and natural resource usePLOS ONE

Dear Dr. Rakotonarivo,

Thank you for submitting your manuscript to PLOS ONE. After careful consideration, we feel that it has merit but does not fully meet PLOS ONE’s publication criteria as it currently stands. Therefore, we invite you to submit a revised version of the manuscript that addresses the points raised during the review process.

**The reviewers have identified several limitations in your study, including the lack of quantitative data, limited scope, potential biases, and the need for deeper analysis. They suggest addressing these concerns by expanding the discussion on generalizability, incorporating external validation, providing a more systematic analysis of heterogeneous patterns in households' responses, and reflecting on ethical considerations and methodological limitations.**

We look forward to receiving your revised manuscript.

Kind regards,

Amar Razzaq, PhD

Academic Editor

PLOS ONE

Journal Requirements:

This work was funded by the USAID Partnerships for Enhanced Engagement in Research (PEER) programme (AID-OAA-A-11-00012 awarded to OSR). OSR was also funded by the European Union (Grant no. DCI-PANAF/2020/420-028), through the African Research Initiative for Scientific Excellence (ARISE) pilot programme implemented by the African Academy of Sciences with support from the European Commission and the African Union Commission. We thank the villagers of Mandena and Andrapengy as well as Randall Kramer, Alexander Duthie and Andrew Bell for comments on an early draft of this manuscript.

This work was funded by the USAID Partnerships for Enhanced Engagement in Research (PEER) programme (AID-OAA-A-11-00012). OSR was also funded by the European Union (Grant no. DCI-PANAF/2020/420-028), through the African Research Initiative for Scientific Excellence (ARISE) pilot programme implemented by the African Academy of Sciences with support from the European Commission and the African Union Commission. We thank the villagers of Mandena and Andrapengy as well as Randall Kramer, Alexander Duthie and Andrew Bell for comments on an early draft of this manuscript.

This work was funded by the USAID Partnerships for Enhanced Engagement in Research (PEER) programme (AID-OAA-A-11-00012 awarded to OSR). OSR was also funded by the European Union (Grant no. DCI-PANAF/2020/420-028), through the African Research Initiative for Scientific Excellence (ARISE) pilot programme implemented by the African Academy of Sciences with support from the European Commission and the African Union Commission. We thank the villagers of Mandena and Andrapengy as well as Randall Kramer, Alexander Duthie and Andrew Bell for comments on an early draft of this manuscript.

4. We note that this data set consists of interview transcripts. Can you please confirm that all participants gave consent for interview transcript to be published?

If they DID provide consent for these transcripts to be published, please also confirm that the transcripts do not contain any potentially identifying information (or let us know if the participants consented to having their personal details published and made publicly available). We consider the following details to be identifying information:

- Names, nicknames, and initials

- Age more specific than round numbers

- GPS coordinates, physical addresses, IP addresses, email addresses

- Information in small sample sizes (e.g. 40 students from X class in X year at X university)

- Specific dates (e.g. visit dates, interview dates)

- ID numbers

Or, if the participants DID NOT provide consent for these transcripts to be published:

- Provide a de-identified version of the data or excerpts of interview responses

- Provide information regarding how these transcripts can be accessed by researchers who meet the criteria for access to confidential data, including:

a) the grounds for restriction

b) the name of the ethics committee, Institutional Review Board, or third-party organization that is imposing sharing restrictions on the data

c) a non-author, institutional point of contact that is able to field data access queries, in the interest of maintaining long-term data accessibility.

d) Any relevant data set names, URLs, DOIs, etc. that an independent researcher would need in order to request your minimal data set.

For further information on sharing data that contains sensitive participant information, please see: https://journals.plos.org/plosone/s/data-availability#loc-human-research-participant-data-and-other-sensitive-data

If there are ethical, legal, or third-party restrictions upon your dataset, you must provide all of the following details (https://journals.plos.org/plosone/s/data-availability#loc-acceptable-data-access-restrictions):

a. A complete description of the dataset

b. The nature of the restrictions upon the data (ethical, legal, or owned by a third party) and the reasoning behind them

c. The full name of the body imposing the restrictions upon your dataset (ethics committee, institution, data access committee, etc)

d. If the data are owned by a third party, confirmation of whether the authors received any special privileges in accessing the data that other researchers would not have

e. Direct, non-author contact information (preferably email) for the body imposing the restrictions upon the data, to which data access requests can be sent

6. We note that Figure 1 in your submission contain map images which may be copyrighted. All PLOS content is published under the Creative Commons Attribution License (CC BY 4.0), which means that the manuscript, images, and Supporting Information files will be freely available online, and any third party is permitted to access, download, copy, distribute, and use these materials in any way, even commercially, with proper attribution. For these reasons, we cannot publish previously copyrighted maps or satellite images created using proprietary data, such as Google software (Google Maps, Street View, and Earth). For more information, see our copyright guidelines: http://journals.plos.org/plosone/s/licenses-and-copyright.

We require you to either present written permission from the copyright holder to publish these figures specifically under the CC BY 4.0 license, or remove the figures from your submission:

Reviewers' comments:

Reviewer's Responses to Questions

**Comments to the Author**

1. Is the manuscript technically sound, and do the data support the conclusions?

Reviewer #1: No

Reviewer #2: No

2. Has the statistical analysis been performed appropriately and rigorously? 

Reviewer #1: No

Reviewer #2: No

3. Have the authors made all data underlying the findings in their manuscript fully available?

Reviewer #1: No

Reviewer #2: No

4. Is the manuscript presented in an intelligible fashion and written in standard English?

Reviewer #1: Yes

Reviewer #2: Yes

5. Review Comments to the Author

**Reviewer #1:** The paper studies the effects of individual and compounding crises (COVID-19, cyclone and vanilla price collapse) on smallholder vanilla farmers and local coping strategies in Madagascar. Applying semi-structured and scenario-based interviews across two case study villages with contrasting enforcement of forest regulations, authors found that the impact of the pandemic, combined with the cyclone event, disrupted livelihoods, resulting in income losses and food security challenges that exacerbated farmer vulnerabilities. The study draw attention to rural vulnerabilities in face of dual shocks of the covid19 and cyclone exposure.

Overall, the topic is important and the research question is valuable. However, I think the paper lacks systematic analysis to draw general conclusions about the impacts of dual shocks in local regions and the effectiveness of coping strategy through local forest exploitation in response to these shocks. In my opinion, the paper is more like a long report article in the media instead of a well-designed research article. Maybe the above is an issue of my personal understanding.

The following are more detailed comments:

1. It will be helpful to do more general analysis on the data based on interviewers’ answer. In the current version, authors just simply list quote out the direct answers to the survey questions. It is not clear how representative one person’s answer is in the aspect of the whole village or the local regions. Or how the surveyed individuals’ answers differ from each other.

2. Related to the first point, it will be helpful to describe the heterogeneous patterns in households’ consumption and income in response to the dual shocks. Are the magnitudes of their income drops the same or different? What factors affect their different responses?

3. How representative are the conclusions from these two villages on the general condition in Madagascar. What the potential application of the shock coping strategy in other regions?

**Reviewer #2: **Lack of Quantitative Data: The study relies solely on qualitative data from semi-structured and scenario-based interviews. While rich in detail, the absence of quantitative data limits the ability to generalize findings and assess the magnitude of impacts.

Limited Geographic Scope: The research is confined to two villages in the SAVA region. This narrow geographic focus may not capture the broader variations and challenges faced by farmers in other parts of Madagascar or other low-income countries.

Sample Size: The sample size of 60 participants (30 per village) is relatively small. This may affect the representativeness of the findings and limit the robustness of the conclusions drawn.

Temporal Scope: Interviews were conducted over a short period (January and February 2023). This limited timeframe may not fully capture the long-term impacts of the crises or seasonal variations in livelihoods and coping strategies.

Potential Bias in Participant Selection: Although participants were randomly selected, the reliance on local authorities and key informants for the sample frame may introduce selection bias, potentially influencing the diversity and representativeness of the sample.

Ethical Considerations: The study obtained verbal consent, which may be less robust than written consent in ensuring participants fully understand the research implications and their rights. This could raise ethical concerns regarding informed consent.

Generalization of Findings: The unique socio-economic and environmental context of the SAVA region, especially its heavy reliance on vanilla farming, may limit the applicability of the findings to other regions with different agricultural and economic profiles.

Absence of Longitudinal Data: The study does not incorporate longitudinal data to track changes over time, which would provide a more comprehensive understanding of the evolving impacts of crises on livelihoods and food security.

Analysis Limitations: While thematic and content analysis were employed, the article does not discuss potential limitations of these methods or how the researchers addressed issues of reliability and validity in the qualitative data analysis.

External Validation: The study lacks a discussion on external validation of the findings. Incorporating perspectives from other regions or cross-referencing with existing quantitative studies could strengthen the credibility and relevance of the conclusions.

Coping Strategies Analysis: The analysis of coping strategies might benefit from a deeper exploration of the effectiveness and sustainability of these strategies. The study does not sufficiently address whether the identified coping mechanisms are viable long-term solutions.

Integration of Local Knowledge: Although the study incorporates local informants, it does not elaborate on how local knowledge and cultural practices were integrated into the analysis. A more explicit discussion of this integration could enhance the contextual relevance of the findings.

Addressing these comments in the revision could provide a more comprehensive and generalizable understanding of the impacts of compounding crises on farmer vulnerability and resilience in Madagascar and similar contexts.

6. PLOS authors have the option to publish the peer review history of their article (what does this mean?). If published, this will include your full peer review and any attached files.

Reviewer #1: No

Reviewer #2: **Yes: **Tsegamariam Dula Sherka

---

## [Author Response · Author response to Decision Letter 0]

24 Jun 2024

Below we responded to the comments made by the two reviewers. Our responses are in blue, addressing all the comments point-by-point, and revised text in the resubmitted manuscript (in red, with line numbers in the manuscript without any tracked changes).

Reviewer #1: The paper studies the effects of individual and compounding crises (COVID-19, cyclone and vanilla price collapse) on smallholder vanilla farmers and local coping strategies in Madagascar. Applying semi-structured and scenario-based interviews across two case study villages with contrasting enforcement of forest regulations, authors found that the impact of the pandemic, combined with the cyclone event, disrupted livelihoods, resulting in income losses and food security challenges that exacerbated farmer vulnerabilities. The study draw attention to rural vulnerabilities in face of dual shocks of the covid19 and cyclone exposure.

Overall, the topic is important and the research question is valuable. However, I think the paper lacks systematic analysis to draw general conclusions about the impacts of dual shocks in local regions and the effectiveness of coping strategy through local forest exploitation in response to these shocks. In my opinion, the paper is more like a long report article in the media instead of a well-designed research article. Maybe the above is an issue of my personal understanding.

Response 1: Many thanks for your succinct summary of the paper and your positive feedback, as well as critiques.

Both reviewers point out that the qualitative nature of the study precludes generalizing to a broader population, a fact with which we agree. The study is intended to be qualitative in nature, obtained through open-answer, semi-structured interviews and scenario-based interviews, which allowed us to glean nuances of individual perceptions of risks, challenges, and adaptive strategies for coping with crises. Although we have included more quantitative evidence to support how often participants reported similar themes (throughout the results section and also in S2 Table), we emphasize that there are risks to over- and mis-interpreting the results if one focuses too narrowly on quantitative results, and these are not to be generalized to larger populations. We also emphasize the importance of even a single report, rather than focus only on majority perceptions. Throughout the manuscript, we made revisions to increase the reporting of empirical findings. There are additions to the methods providing more background on the field of context-based narrative and thematic analyses, and interpretations of our results into this broader field in the discussion. We also point out the limitations, specifically that we should not generalize to broader regions from our results, especially not based on the reported frequency data. Lastly, we edited the language such that we hope the reviewer and readers will not perceive the writing style as media rather than academic, though there should not be such a stark dichotomy.

To give some specifics:

First, we outlined our research aims in lines 68-81 and edited the wording to increase clarity:

This research aims to study the individual and compounding effects of global crises (COVID-19, cyclones and vanilla price collapse) on farmers' lives across two case study villages in northeast Madagascar. Using semi-structured and scenario-based interviews, we identified various mechanisms by which these cascading crises can consecutively and concurrently impact the livelihoods and food security of farmers in these case villages. We further explored the knock-on impacts on natural resource use and biodiversity. Finally, we report the various strategies employed by farmers to cope with these challenges. By focusing on the context-based specific realities of the case study villages, this article contributes to a better understanding of the challenges faced by farmers in this region of Madagascar and provides wider lessons for similar contexts in low-income countries. We explored these research aims using a qualitative approach. Qualitative approaches can help provide detailed, vivid, and context-sensitive descriptions of the data and and gain an in-depth understanding of the context-specific challenges and nuanced perspectives of place-based actors [14]. 

Second, to more clearly describe how we address these research aims, we carefully elaborated on the semi-structured interview guide and scenario-based guide (please see lines 96-113). 

The semi-structured interview guides (S1 File) consisted of 18 questions that covered participant basic demographics, including age, household size, education level, and income sources. These interviews also examined the impacts of various crises, such as cyclones, COVID-19, and fluctuations in vanilla prices on livelihoods and food security. Additionally, participants were asked about the coping strategies they adopted in response to each crisis and the potential effects of each crisis and their compounding effects on the use of natural resources. 

The scenario-based interview guide (S2 File) covered six hypothetical scenarios encompassing the status quo, an increase or decrease in vanilla prices, land degradation, diseases affecting vanilla cultivation, the restrictions during COVID-19, and extreme weather events such as cyclones. For each scenario, participants were prompted to share the anticipated effects of these crises on their lives, and their coping strategies. This method allowed us to delve into the nuanced decision-making processes of farmers, including a long-term perspective on adaptive responses they have used and their perspectives in the face of uncertainty. Respondents were prompted to speak on any of the crises they chose to discuss, without restricting the timeframe. 

Third, we analyzed these data using both thematic and content analysis methods. Thematic analysis in qualitative research provides a rich, detailed, complex account of data. As Braun and Clarke [1] put it, thematic analysis is a method for “identifying, analyzing and reporting patterns (themes) within data. It minimally organizes and describes the data set in (rich) detail.” Thematic analysis involves identifying and understanding key themes in the data and how they relate to one another. Eponymously, the themes derived from the data actively construct the patterns of meaning to address our research aims. In short, themes are ‘a patterned response or meaning’ derived from coded data that represent overarching ideas embedded within the larger data set. 

We have added more details in lines 127-150, and also suggested how our results should be interpreted:

We analyzed the data using thematic and content analysis methods [16,17]. Thematic analysis involved identifying and analyzing themes in the data, including initial coding of meaningful units, searching for common themes, revising and defining themes, and interpreting the results [16]. We specifically searched for any themes and patterns indicating the potential effects of various crises, including COVID-19, cyclones, and vanilla price fluctuations on household livelihoods and food insecurity, the effects of these crises on the use of natural resources and biodiversity, and the diverse strategies adopted by farmers. The aim is to uncover and understand key themes in the data and how they relate to one another. The content analysis involved a systematic analysis of the key themes identified from the thematic analysis [17]. It focuses on the reoccurrence of themes at a more surface-level of analysis i.e. their frequency. 

Researchers and readers should use caution when interpreting such frequencies, however [18]. Frequencies may help get a better feel for how often themes occur within the specific sample, and advance qualitative research within a discipline that are not familiar with or oriented to qualitative research. Nevertheless, incorporating frequencies in qualitative research may misrepresent data as not every interviewee is asked the same questions in the same way [19]. Number can also detract from the more valuable, detailed, and nuanced insights uncovered from qualitative research [19]. Similarly, though a theme or response that only occurs once may seem unimportant based on frequency of occurrence, it may represent the unique challenge, solution, idea, or theory that could influence social, anthropological, and policy change. Lastly, readers should not interpret frequencies as generalizations to broader populations; instead, one should assess the transferability of the findings, sometimes also referred to as analytic generalization or case-to-case generalization [18,19]. 

Fourth, to explain the systematic nature of our analysis and the rigor with which the data were treated, we added a paragraph in lines 151-160 to emphasize our efforts to ensure the final analytic framework represents a credible account of the data.

Coding was conducted iteratively by HRH to generate codes and themes that summarized and captured the essence of the responses. a second coder (OSR) used the draft coding scheme to independently code five randomly selected sample transcripts. To ensure consistency in coding and mitigate the subjectivity of a single coder, we calculated the intercoder reliability, the Cohen’s Kappa coefficient which is a numerical measure of the agreement between different coders regarding how the same data should be coded [20]. We considered a coefficient with a value of 0.70 or higher acceptable. Codes that did not meet this threshold were jointly reviewed, and conflicting interpretations were resolved by either merging codes or refining code definitions. These analyses were conducted with NVivo 12 software.

The following are more detailed comments:

1. It will be helpful to do more general analysis on the data based on interviewers’ answer. In the current version, authors just simply list quote out the direct answers to the survey questions. It is not clear how representative one person’s answer is in the aspect of the whole village or the local regions. Or how the surveyed individuals’ answers differ from each other.

Response 2: We thank the reviewer for this suggestion, and have taken the following steps. To provide the readers with more quantitative evidence, a table is included in the supplementary materials (Table S2), synthesizing the frequency or occurrence of specific themes. We present these results as descriptive statistics and emphasize caution in reading into significance or representativeness as in quantitative methods. A theme that only appeared once doesn’t necessarily indicate that it is less important.

As described above, we provided more details on the thematic and content analyses we conducted to address our research questions. The purpose of these qualitative approaches is to provide in-depth explanations and meanings rather than generalizing findings. Our paper provides detailed, vivid, and context-sensitive descriptions of the data in our manuscript, which is essential in qualitative research. The verbatim quotes from the transcripts not only provide evidence of the general themes that respondents reported, but deep, rich, and context-specific evidence, both of which support our findings.

In the methods, we provide more background on the following papers to further develop the theory, and describe how frequencies can complement and enhance narratives and improve transparency (see lines 138-147). We especially build on the works of Black 1994; Sandelowski 2001; Neale et al 2014; Maxwell 2010; Hannah and Lautsch 2011 (references provided at the end). More specifically, frequencies may help get a better feel for how often themes occur, and advance qualitative research within a discipline that are not familiar with or oriented to qualitative research. Nevertheless, incorporating frequencies numbers in qualitative research may mislead readers or replace the actual description of evidence. 

To be more specific in this response, and as summarized in the methods of the revised manuscript, Maxwell (2010, p. 479) articulated the potential pitfalls of overinterpreting frequency data derived from qualitative research:

Numbers can lead to the inference (by either the researcher or the audience) of greater generality for the conclusions than is justified, by slighting the specific context within which this conclusion is drawn. A particular setting or sample may be unrepresentative, and a facile reading of quantitative results may lead a reader to ignore this limitation. Qualitative research is intrinsically local, and any claims for the generality of its conclusions rely on a different kind of argument from that of quantitative research, what Yin (1994) called analytic rather than statistical generalization.

Generalizations can be made from qualitative research, but just not in the same way as quantitative results are (which aims to generalize findings to a larger population). The type of generalization that is applicable to our qualitative analysis is transferability which is sometimes also referred to as analytic generalization or case-to-case generalization. The question is “to what extent are these results transferable to other settings?’ For example, a development practitioner who is designing a livelihood intervention might want to know: “Are these patterns also true in the context I work with and what would be the most effective policy levers increasing local resilience?” When readers feel as though this can be the case – when they believe that research overlaps with their own situation and/or they can intuitively transfer the findings to their own action –, then the research can be said to generalize through transferability.

We emphasize this in our revised manuscript both in the methods and in the policy implications:

Lines 147-150

Lastly, readers should not interpret frequencies as generalizations to broader populations; instead, one should assess the transferability of the findings, sometimes also referred to as analytic generalization or case-to-case generalization.

Lines 482-485:

Given the relatively small sample and our qualitative approaches, we cannot provide generalizations to the population from which the sample was drawn. Our research findings can however be transferable to other settings and provide wider lessons for similar contexts in low-income countries.

Throughout the discussion, we shed light on the transferability of our results by reflecting on how our results compare with findings in similar contexts. For instance:

Lines 409-415:

The impacts of government-imposed lockdowns and restrictions on livelihoods were severe, affecting nearly 70% of households. Similar magnitudes of impact were observed in other parts of Madagascar [22] as well as among Ugandan households, where two-thirds reported income losses during the pandemic [23]. Farmers encountered challenges due to travel restrictions that hindered their ability to sell produce freely, resulting in limited market access and reduced income [23] . 

Lines 449-456

Simultaneous shocks can magnify vulnerabilities, leading to cascading impacts on livelihoods and well-being. These findings are in line with previous studies reporting that the pandemic posed new challenges to livelihood strategies while complicating disaster management during cyclone season [29,30]. The intersection of the pandemic and cyclone Amphan in Bangladesh highlighted the need to consider complex temporalities of disasters [31]. These findings are also consistent with similar studies documenting the compounding impacts of multiple crises on vulnerable populations [32,33].

Lines 485-496

Our study warrants the need for targeted support systems that integrate disaster risk reduction, livelihood diversification, and market resilience strategies to foster long-term sustainability and strengthen the resilience of households facing multiple crises. 

Given the vulnerability of households heavily dependent on single cash crops like vanilla, policymakers should promote diversification of income sources and agricultural practices as recommended in other similar contexts [34,35]. In addition, disaster preparedness and risk reduction efforts can be strengthened through policy in

---

## [Decision Letter · Decision Letter 1]

17 Sep 2024

Compounding impacts of COVID-19, cyclone and price crash on vanilla farmers’ food security and natural resource use

PONE-D-24-09752R1

Dear Dr. Rakotonarivo,

We’re pleased to inform you that your manuscript has been judged scientifically suitable for publication and will be formally accepted for publication once it meets all outstanding technical requirements.

Kind regards,

Amar Razzaq, PhD

Academic Editor

PLOS ONE

Additional Editor Comments (optional):

Reviewers' comments:

Reviewer's Responses to Questions

**Comments to the Author**

1. If the authors have adequately addressed your comments raised in a previous round of review and you feel that this manuscript is now acceptable for publication, you may indicate that here to bypass the “Comments to the Author” section, enter your conflict of interest statement in the “Confidential to Editor” section, and submit your "Accept" recommendation.

Reviewer #1: All comments have been addressed

Reviewer #3: All comments have been addressed

2. Is the manuscript technically sound, and do the data support the conclusions?

Reviewer #1: Yes

Reviewer #3: Yes

3. Has the statistical analysis been performed appropriately and rigorously? 

Reviewer #1: Yes

Reviewer #3: Yes

4. Have the authors made all data underlying the findings in their manuscript fully available?

Reviewer #1: Yes

Reviewer #3: Yes

5. Is the manuscript presented in an intelligible fashion and written in standard English?

Reviewer #1: Yes

Reviewer #3: Yes

6. Review Comments to the Author

Reviewer #1: Overall, the topic of the paper is important and the research question is valuable. From the typical standard of an analytical paper, the paper is slightly weak in quantitative analysis. While mainly using qualitative approaches, the paper suffers from the small sample size and external validity of a specific study regions.

However, the authors have addressed my previous questions reasonably and thoroughly. I recommend acceptance of the paper.

Reviewer #3: The Authors have addressed all the review feedback and justified it properly. Wherever changes are required, they have incorporated them as well.the paper has improved and in a publishable format as well

7. PLOS authors have the option to publish the peer review history of their article (what does this mean?). If published, this will include your full peer review and any attached files.

Reviewer #1: No

Reviewer #3: No

---

## [Editor Report · Acceptance letter]

24 Sep 2024

PONE-D-24-09752R1 

PLOS ONE

Dear Dr. Rakotonarivo, 

I'm pleased to inform you that your manuscript has been deemed suitable for publication in PLOS ONE. Congratulations! Your manuscript is now being handed over to our production team.

Kind regards, 

on behalf of

Associate Professor Amar Razzaq 

Academic Editor

PLOS ONE